# The transcription factor HIF-1α in NKp46+ ILCs limits chronic intestinal inflammation and fibrosis

Eric Nelius[1], Zheng Fan[1], Michal Sobecki[1], Ewelina Krzywinska[1], Shunmugam Nagarajan[1], Irina Ferapontova[1], Dagmar Gotthardt[2], Norihiko Takeda[3], Veronika Sexl[4], Christian Stockmann[1,5,6]

Innate lymphoid cells (ILCs) are critical for intestinal adaptation to microenvironmental challenges, and the gut mucosa is characterized by low oxygen. Adaptation to low oxygen is mediated by hypoxia-inducible transcription factors (HIFs), and the HIF-1α subunit shapes an ILC phenotype upon acute colitis that contributes to intestinal damage. However, the impact of HIF signaling in NKp46[+] ILCs in the context of repetitive mucosal damage and chronic inflammation, as it typically occurs during inflammatory bowel disease, is unknown. In chronic colitis, mice lacking the HIF-1α isoform in NKp46+ ILCs show a decrease in NKp46[+] ILC1s but a concomitant rise in neutrophils and Ly6C^high macrophages. Single-nucleus RNA sequencing suggests enhanced interaction of mesenchymal cells with other cell compartments in the colon of HIF-1α KO mice and a loss of mucus-producing enterocytes and intestinal stem cells. This was, furthermore, associated with increased bone morphogenetic pathway–integrin signaling, expansion of fibroblast subsets, and intestinal fibrosis. In summary, this suggests that HIF-1α–mediated ILC1 activation, although detrimental upon acute colitis, protects against excessive inflammation and fibrosis during chronic intestinal damage.

## Introduction

Innate lymphoid cells (ILCs) are a heterogeneous population of non-B, non-T lymphocytes with a crucial role in intestinal adaptation to microenvironmental challenges (Colonna, 2018; Guendel et al, 2020). Therefore, ILCs play a critical role during intestinal damage and inflammation (Diefenbach et al, 2014; Eberl et al, 2015; Klose & Artis, 2016). The three major groups of ILCs are defined by the secretion of distinct cytokines, the expression of lineage-defining transcription factors that define the phenotype, and distinctive surface markers. T-bet+ ILC1s, which include the natural killer (NK) cell subset, produce proinflammatory IFN-γ; GATA3+ ILC2s secrete IL-5, IL-9, IL-13, and amphiregulin; and RORγt+ ILC3s produce IL-22 and IL-17. In addition to NKp46-ILC3s, murine RORγt+ ILC3s comprise a NKp46+ ILC3 subset that is specialized in IL-22 secretion, which induces the expression of prohomeostatic factors in gut epithelial cells to maintain barrier integrity and homeostasis (Sanos et al, 2011; Diefenbach et al, 2014; Eberl et al, 2015; Klose & Artis, 2016). The mucosa of the large intestine contains NKp46[+] IL-22–producing group 3 ILCs (ILC3s) and IFN-γ–producing group ILCs (ILC1s) (Bernink et al, 2013). The mucosa is characterized by physiological hypoxia, often termed "physoxia," and cellular adaptation to low oxygen is mediated by hypoxia-inducible transcription factors (HIFs) (Manresa & Taylor, 2017). HIFs, with HIF-1 and HIF-2 being the most extensively studied isoforms (Kaelin & Ratcliffe, 2008; Semenza, 2012), are basic helix–loop–helix transcription factors that consist of a constitutively expressed β-subunit and an oxygen-labile α-subunit. In the presence of oxygen, the α-subunit is hydroxylated by prolyl hydroxylases and subsequently degraded through the ubiquitin–proteasome pathway via interaction with its negative regulator von Hippel–Lindau protein (Schofield & Ratcliffe, 2004; Kaelin & Ratcliffe, 2008). Multiple studies have shown that HIF activation in intestinal epithelial cells can protect from colitis (Cummins et al, 2008; Tambuwala et al, 2010, 2015). We have previously demonstrated that the colon of mice with HIF-1α deficiency in NKp46+ cells contains twice as many NKp46+ ILC3s, whereas NKp46+ ILC1s, including NK cells, were comparable to WT littermates, resulting in a significant increase in ILC3/ILC1 ratio. The relative abundance of ILC3s over ILC1s was associated with a higher fraction of IL-22–expressing ILC3 NKp46+ cells, decreased frequencies of IFN-γ–expressing ILC1s, and augmented expression of gut-homeostatic genes such as *Reg3b*, *Reg3c*, *Defa21*, *Muc2*, and *Muc5* in the colon (Krzywinska et al, 2022). Upon acute experimental colitis, mucosal hypoxia is further aggravated, and we have demonstrated that HIF-1 fosters an ILC1 phenotype, which contributes to intestinal damage, whereas HIF-1α deficiency in NKp46+ cells protects from acute experimental colitis (Krzywinska et al, 2022). This is associated with an increased ILC3-to-ILC1 ratio in the colon, with no genotype-specific differences among

[1]Institute of Anatomy, University of Zurich, Zurich, Switzerland   [2]Institute of Pharmacology and Toxicology, University of Veterinary Medicine, Vienna, Austria   [3]Division of Cardiology and Metabolism, Center for Molecular Medicine, Jichi Medical University, Shimotsuke, Japan   [4]University of Innsbruck, Innsbruck, Austria   [5]Comprehensive Cancer Center Zurich, Zurich, Switzerland   [6]Zurich Kidney Center, Zurich, Switzerland

Correspondence: christian.stockmann@anatomy.uzh.ch

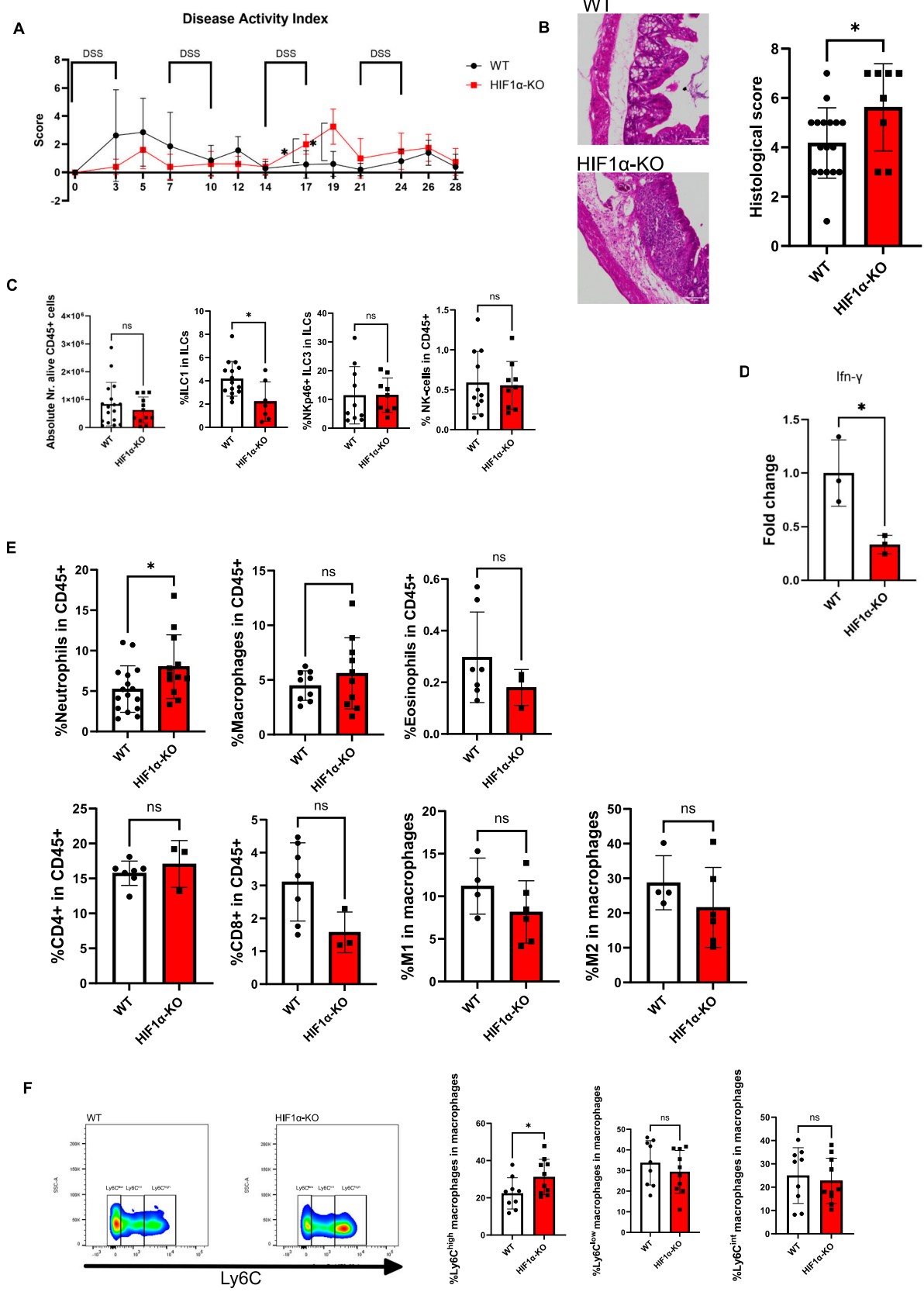

other immune cell subsets, as well as the reduced expression of the colitogenic ILC1 cytokine *Ifng-γ* (Krzywinska et al, 2022).

Of note, inflammatory bowel diseases, including colitis, are characterized by chronic inflammation with repetitive mucosal damage during flare-ups (Abraham & Cho, 2009; Rieder & Fiocchi, 2009), along with the development of intestinal fibrosis (Rieder et, 2017). However, the impact of HIF signaling in NKp46+ ILCs on gut homeostasis and repair during repetitive intestinal damage and chronic inflammation has not been studied. We found that in a model of chronic colitis, mice lacking the HIF-1α isoform in NKp46+ ILCs show a decrease in NKp46+ ILC1s in the colon mucosa, which resembles the phenotype during acute colitis. Yet, in the chronic colitis setting, the absence of HIF-1α in NKp46+ ILCs leads to an increased disease activity index and a concomitant rise in neutrophils and a Ly6C$^{high}$ macrophage subset that is considered as profibrotic (Ramachandran et al, 2012). Single-cell RNA sequencing revealed a loss of mucus-producing enterocytes and an enhanced interaction of mesenchymal cells with other cell compartments in the colon of HIF-1α KO mice. This was, furthermore, associated with an expansion of profibrotic fibroblast subsets along with increased collagen deposition and intestinal fibrosis. In summary, this suggests that HIF-1α–mediated ILC1 activation, although detrimental upon acute colitis, prevents against excessive inflammation and fibrosis during chronic intestinal damage.

# Results

## HIF-1α in NKp46+ cells limits inflammation and the severity of chronic colitis

In order to test the role of HIF-1α in NKp46+ gut ILCs in chronic injury in the large intestine, we used mice with a targeted deletion of HIF-1α, via crosses of the loxP-flanked HIF-1α allele to the *Ncr1* (NKp46) promoter–driven Cre recombinase, specific to NKp46-expressing ILC1s, including NK cells and ILC3s (Ncr1$^{cre+}$ HIF-1α$^{fl+/fl+}$ mice, termed HIF-1α KO) (Eckelhart et al, 2011; Krzywinska et al, 2017) and WT littermates to a scheme of repetitive dextran sodium sulfate (DSS) exposure (Fig 1A), which induces mucosal damage and chronic colitis (Okayasu et al, 1990). Surprisingly, HIF-1α KO mice exhibited a transiently enhanced disease activity that comprises weight loss, stool consistency, and hematochezia (Friedman et al, 2009) (Fig 1A) during the third DSS challenge and a worsened histological score at the endpoint (Fig 1B), despite a decrease in ILC1 frequencies (Fig 1C). Of note, the total number of CD45+ cells, the frequencies of NKp46+

ILC3, NK cells, and CD4+ and CD8+ T cells as assessed by flow cytometry were similar across genotypes (Fig 1C and E; for gating strategy, please see Fig S1A and B). Along with the decrease in ILC1 frequencies, we found a decrease in the mRNA expression of the ILC1 signature cytokine Ifng in the colon of HIF-1α KO mice (Fig 1D). Interestingly, within the myeloid cell compartment, HIF-1α KO mice experienced increased colonic neutrophil infiltration, whereas eosinophil and macrophage frequencies including the M1 and M2 subsets were comparable to WT mice (Fig 1E). In addition to the M1/M2 classification, macrophage phenotypes can be delineated by expression levels of the surface markers Ly6C (Ramachandran et al, 2012). Of note, HIF-1α deficiency in NKp46+ cells results in an increase in Ly6C$^{high}$ macrophages (Fig 1F). In summary, this suggests that similar to the acute colitis setting, the deletion of HIF-1α in NKp46+ cells leads to the relative depletion of tissue-resident ILC1s from the colon (Krzywinska et al, 2022), yet upon chronic mucosal injury, this is associated with a concomitant increase in neutrophils and Ly6C$^{high}$ macrophages from the circulation. Therefore, we concluded that in contrast to the setting of acute colitis (Krzywinska et al, 2022), loss of HIF-1α in NKp46+ upon chronic intestinal injury promotes inflammation and mucosal damage.

## HIF-1α deficiency in NKp46+ cells deviates the gene regulatory network upon chronic colitis

The colon comprises various epithelial and mesenchymal cell types that form a functional circuit with the immune cell compartment (Neurath, 2019). Given the altered inflammatory response in the colon, we wanted to assess the impact of HIF-1α deficiency in NKp46+ cells on the gut microenvironment beyond the immune cell compartment at the whole tissue level and at single-cell resolution. To this end, we performed single-nucleus RNA sequencing on frozen colon tissues from WT and HIF-1α KO mice after chronic DSS exposure. We analyzed a total of 9,668 nuclei and detected a median of 1,277 gene transcripts per cell in WT and 2,182 in HIF-1α KO mice. Supervised clustering readily allowed to distinguish several clusters, including absorptive and mucus-producing enterocytes (enterocyte Abs and enterocyte MP, respectively), goblet cells based on transcript signatures (Fig S2A), enteroendocrine cells and intestinal stem cells, along with endothelial cell subsets and the mesenchymal compartment (Fig 2A). A comparison of the different clusters across genotypes revealed the relative depletion of mucus-producing enterocytes and intestinal stem cells in the colon of HIF-1α KO (Fig 2B). This observation is further supported by immunostainings for LGR5 (a marker for intestinal stem cells) (Barker et al, 2007) and mucin 2 (MUC2, a marker for mucus-producing

---

**Figure 1. Deletion of HIF-1α in NKp46+ cells leads to a proinflammatory phenotype in the proximal colon upon chronic dextran sodium sulfate exposure.**
**(A)** Disease activity index of HIF-1α KO compared with WT mice. Mice were treated with 2% dextran sodium sulfate in drinking water four times for 3 d with the recovery phase of 4 d; n WT = 5, n KO = 4; data are mean values ± SD; *P < 0.05, two-tailed *t* test. **(B)** Representative images of H&E-stained proximal colon and analysis of the histological score of inflammation. Pooled data of three experiments; n WT = 17, n KO = 8; data are mean values ± SD; *P < 0.05, two-tailed *t* test. **(C)** Absolute number of CD45+ cells and abundance of different innate lymphoid cell (ILC) subsets in % of alive CD45+ cells at the endpoint. ILC1s defined as alive CD45+, Lin negative (CD11c, CD19, Ly6G, Ter119, TCR-*β*, TCR-*γ/δ*), CD127+, RORγt negative, NKp46+, T-bet+, CD49a+, NK1.1+. NK cells as alive CD45+, Lin negative, CD127 negative, NKp46+, Eomes+, NK1.1+. ILC3 as alive CD45+, Lin negative, CD127+, RORγt+, NKp46+. Pooled data of two independent experiments; n WT ≥ 10, n KO ≥ 7; data are mean values ± SD; *P < 0.05, two-tailed *t* test. **(D)** Fold change of the IFN-γ mRNA expression of whole colon of WT and HIF-1α KO mice. Data were normalized to *16s*. n WT = 3, n KO = 3. Data are mean values ± SD; *P < 0.05, two-tailed *t* test. Two outliers have been excluded using the ROUT method (Q = 1%). **(E)** Flow cytometric analysis of alive CD45+, neutrophils, macrophages, eosinophils, CD4+ and CD8+ T cells, and M1 (iNOS+) and M2 (CD206+) macrophages at the endpoint. Pooled data of two experiments; n KO ≥ 3, n WT ≥ 4; data are mean values ± SD; *P < 0.05, two-tailed *t* test. **(F)** (Left) Representative flow cytometric images of Ly6C+ gated on macrophages (alive CD45+, CD11b+, F4/80+) in HIF-1α WT and KO in NKp46+ cells and results (right) of % low/intermediate/high Ly6C. Pooled data of two experiments; n KO = 10, n WT = 9; data are mean values ± SD; *P < 0.05, two-tailed *t* test.

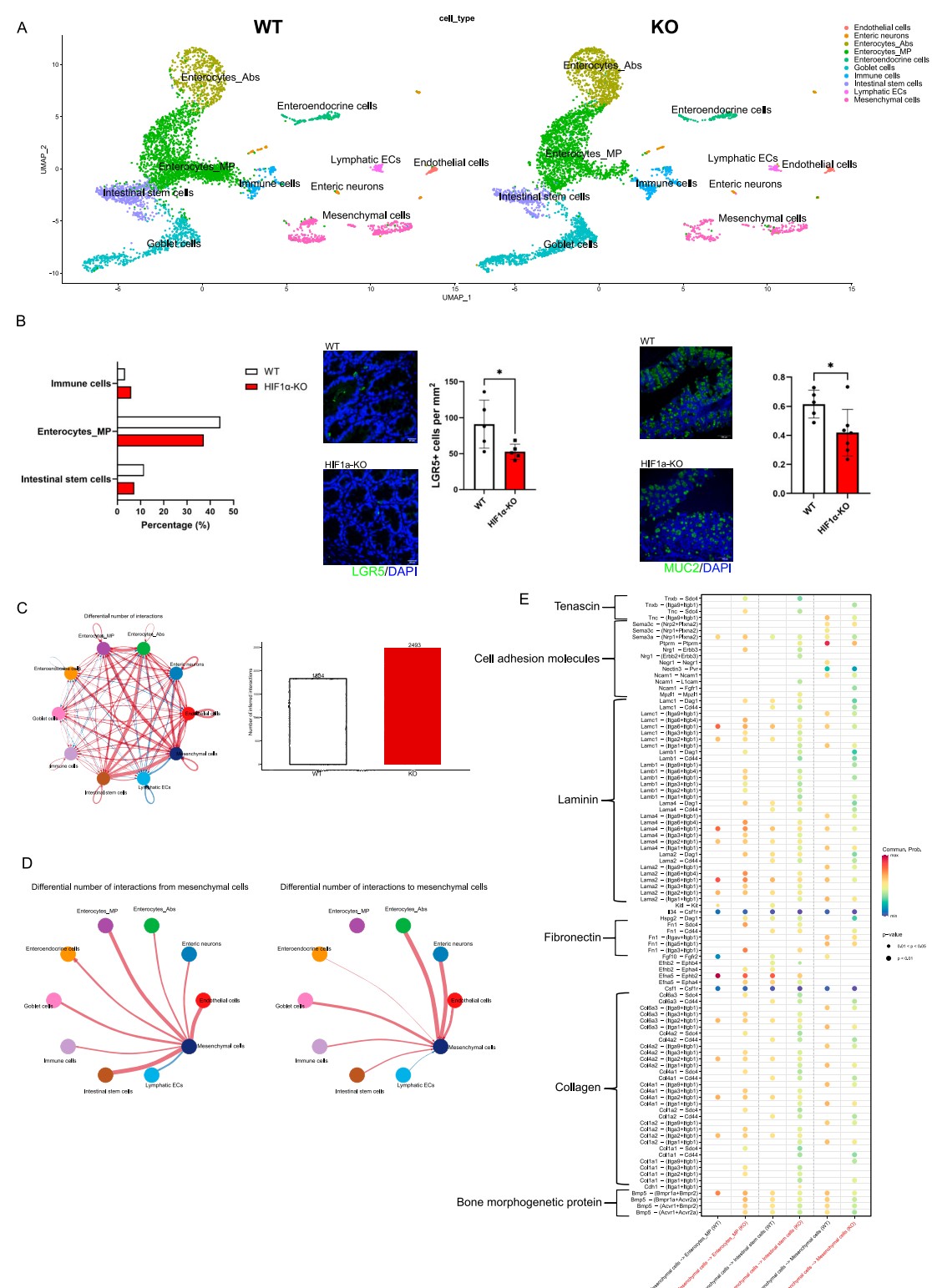

**Figure 2.  Single-nucleus RNA sequencing reveals a proinflammatory phenotype of HIF-1α KO mice with increased overall signaling.**
**(A)** UMAP plots and corresponding cell types of the whole colon tissue at the endpoint of WT and HIF-1α KO in NKp46+ cells. A total of 9,668 nuclei were analyzed. **(B)** Left: distribution of immune cells, mucus-producing enterocytes, and intestinal stem cells. Middle: representative images of immunofluorescent LGR5 staining (green) and DAPI (blue) on colon sections. Data were analyzed and calculated (cells per mm²). Right: representative images of immunofluorescent MUC2 staining (green) and DAPI (blue) on colon sections. Pooled data of two experiments; n KO ≥ 5, n WT = 5; data are mean values ± SD; *$P < 0.05$, two-tailed $t$ test. **(C)** CellChat analysis of all cell types in the colon. Red arrows show increased, and blue arrows show decreased, signaling in the colon of HIF-1α KO. **(D)** Number of interactions from and toward mesenchymal

goblet cells and mucus-producing enterocytes) (Pelaseyed et al, 2014) on colon sections from WT and HIF-1α KO mice after chronic DSS exposure at the endpoint (day 28). As shown in Fig 2B, the colon of HIF-1α KO mice exhibits a reduction in LGR5+ intestinal stem cells and the area of MUC2+ cells. Of note, MUC-2 is expressed by goblet cells and mucus-producing enterocytes and therefore does not allow to distinguish the two cell types. However, given that the number of goblet cells was not reduced according to the single-cell RNA-sequencing analysis, these data indicate a reduction in mucus-producing cells, including enterocytes, in the colon of HIF-1α KO mice.

Next, we interrogated and compared the manifold putative communication pathways between the different non-immune cell compartments in a genotype-specific manner (Fig 2C, brown arrows depict enhanced, and blue arrows depict decreased, interactions in HIF-1α KO mice, respectively) using CellChat (Jin et al, 2021). We observed an increased number of inferred interactions in the absence of HIF-1α in NKp46+ cells (Fig 2C), and the frequency of interactions in the colon of HIF-1α KO mice involved particularly mesenchymal cells and endothelial cells, as well as enterocytes and intestinal stem cells in the epithelial cell compartment (Fig 2C, brown arrows for enhanced, and blue arrows for decreased, interactions in the colon HIF-1α KO mice, respectively). In the absence of HIF-1α in NKp46+ cells, the mesenchymal cells showed increased interactions with endothelial cells, as well as the intestinal stem cells and the enterocyte MP (Fig 2C and D), with the two latter ones being relatively depleted in the colon HIF-1α KO mice (Fig 2A). Therefore, we next interrogated the signaling pathways between those compartments in a genotype-specific manner. The enhanced signaling pathways from mesenchymal cells to intestinal stem cells and the enterocyte MP in the colon of HIF-1α KO mice dominantly comprised ligands and receptors of the bone morphogenetic pathway (BMP) and integrin signaling (Fig 2E). In addition, mesenchymal-to-mesenchymal cell communication was increased via CD44 signaling in the colon of HIF-1α KO mice. Of note, CD44 has been shown to promote organ fibrosis (Chen et al, 2017; Osawa et al, 2021) (Fig 2E). Therefore, we concluded that loss of HIF-1α in NKp46+ cells promotes aberrant signaling circuits involving BMP, as well as integrin signaling and the mesenchymal cell compartment as a major signaling hub, which eventually contributes to aggravated chronic colitis (Fig 1).

## HIF-1α in NKp46+ cells limits myofibroblast expansion and intestinal fibrosis

Of note, BMP and integrin signaling are involved in the development of tissue fibrosis (Herrera et al, 2017). In the context of inflammatory bowel diseases and chronic colitis, one of the major complications is the development of intestinal fibrosis and reduced gut motility (Baumgart & Sandborn, 2007), owing to the expansion of profibrotic fibroblast subsets that deposit extracellular matrix components including collagens and fibronectin (Andoh et al, 2007). Given the enhanced signaling events toward the mesenchymal cell compartment in HIF-1α KO mice (Fig 2), this prompted us to analyze the single-nucleus RNA sequencing on the mesenchymal

cell compartment across genotypes in a more detailed manner. In general, the mesenchymal cell compartment in the colon from HIF-1α KO mice showed an increased expression of the myofibroblast signature genes, *Acta2* (encoding for alpha-smooth muscle actin), *Des* (Desmin), *Fn1* (Fibronectin 1), *Postn* (Periostin), *Lamb1* (Laminin 1), *Eln* (Elastin), and *Col1a1* (Collagen 1) (Fig S3). Next, clustering of mesenchymal cells across both genotypes revealed 10 distinct clusters based on transcript signatures (Fig 3A and B), with four subclusters in the fibroblast population (termed Fibro 1, 2, 3, and 4) and eight subclusters in the smooth muscle cell subset (termed SMC 1, 2, 3, 4, 5, 6, 7, and 8) (Fig 3A and B). Mesenchymal cells in the colon of WT and HIF-1α KO mice showed a distinct distribution across the fibroblast and SMC clusters with a relative shift from smooth muscle cells toward the fibroblast compartment in HIF-1α KO mice as displayed in Fig 3C. In the colon of WT mice, mesenchymal cells clustered differentially in the fibroblast clusters 1 and 4 (38.7% versus 38.2% and 15.1% versus 25.5% in HIF-1α KO) and the SMC clusters 2 and 5 (26.9% versus 20% and 12% versus 8.2% in HIF-1α KO), whereas in HIF-1α KO mice, mesenchymal cells were predominantly represented in the fibroblast clusters 2 and 3 (19.8% versus 39.6% and 7.5% versus 15.6% in WT) and the SMC clusters 4 and 6 (13.4% versus 14.9% and 6.2% versus 1.3% in WT) (Fig 3C). Of note, the fibroblast cluster 2 and SMC clusters 4 and 6, which are more abundant in the colon of HIF-1α KO mice, are characterized by a high expression of several genes involved in extracellular structure matrix organization and connective tissue development including *Col1a1, Col1a2, Col3a1, Adamts3, Adamts9, Fn1, Postn* (Fig 3B and D). Hence, loss of HIF-1α In NKp46+ cells resulted in the enrichment of potentially fibrogenic cells in the fibroblast and SMC compartment.

To investigate the relationship between these clusters, we performed RNA velocity analysis (La Manno et al, 2018) (Fig 3E). This revealed two main trajectories, along which mesenchymal cells in the colon of WT and HIF-1α KO NKp46+ progressed differentially. Within the fibroblast populations, the trajectory, which was dominant in WT mice, leads through the clusters Fibro 3 toward Fibro 1 and 4. In contrast, the trajectory that showed abundance in HIF-1α KO mice leads to Fibro 2 (Fig 3E). For the SMC compartment, the trajectory in WT mice leads preferentially through the cluster SMC 3 toward SMC 2 and 5, whereas the trajectory in HIF-1α KO mice dominantly leads through the cluster SMC 3 toward SMC 4 and 6 (Fig 3E). Therefore, we concluded that loss of HIF-1α in NKp46+ promotes a microenvironment that facilitates the expansion of fibrogenic fibroblast and SMC subsets. This further corroborated immunostainings for alpha-SMA and PDGFR-alpha on colon sections. As shown in Fig 3F, the submucosa of HIF-1α KO mice showed an increase in the PDGFR-alpha+ area along with increased thickness of the colon wall (Fig 3G). Finally, Sirius Red staining on colon tissues from WT and HIF-1α KO mice at the endpoint after chronic DSS exposure to detect the collagen distribution and the tunica muscularis of HIF-1α KO mice shows increased collagen deposition (Fig 3H). This suggests that the relative abundance of fibrogenic mesenchymal subsets upon loss of HIF-1α in NKp46+ cells facilitates intestinal fibrosis.

---

cells. **(E)** Bubble plots of corresponding ligands–receptors from mesenchymal cells toward the mucus-producing enterocytes, intestinal stem cells, and the mesenchymal cell compartment of WT and HIF-1α KO mice. Pooled samples of n = 3 animals per group at the endpoint.

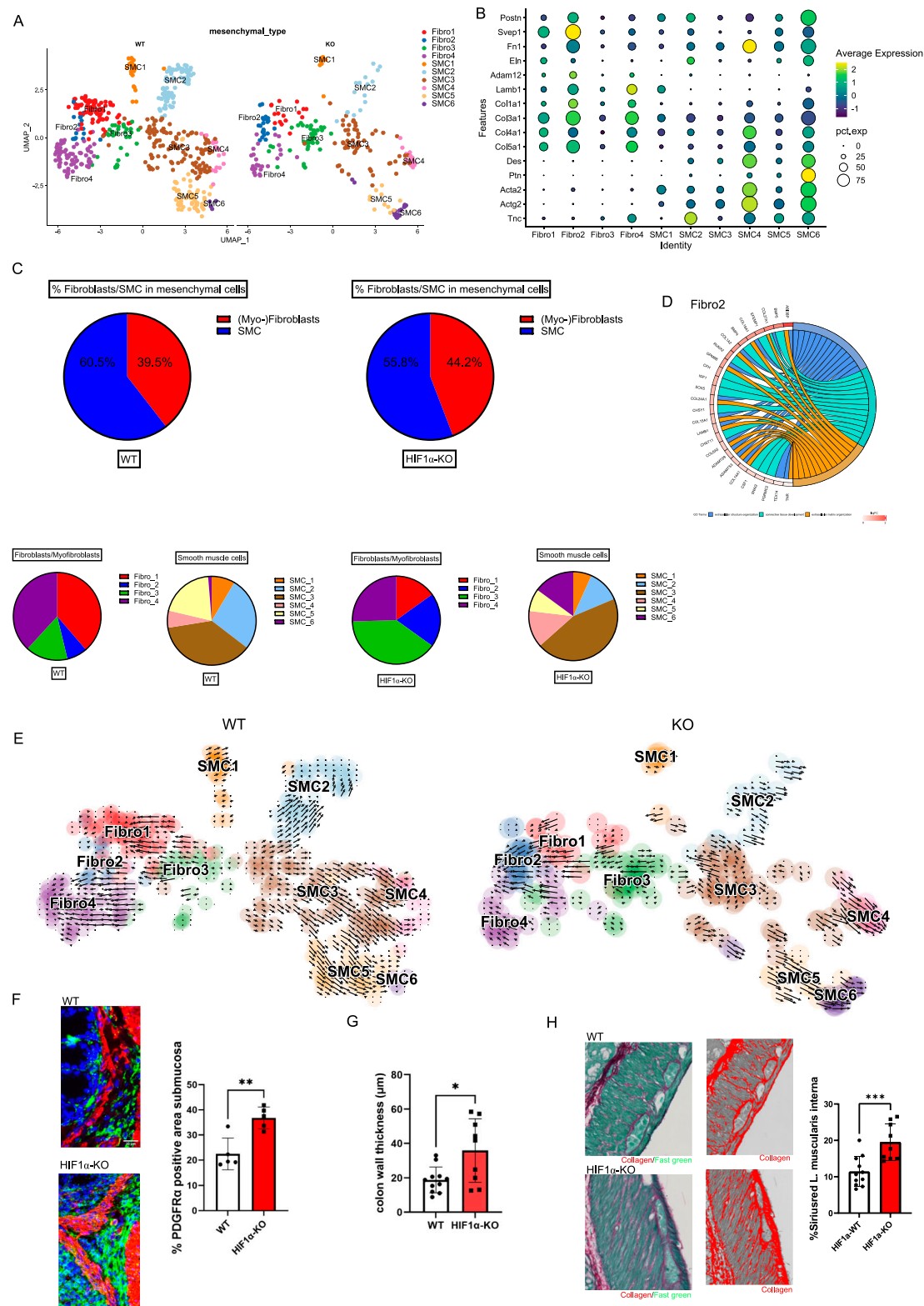

**Figure 3. Single-nucleus RNA sequencing reveals an increased frequency of profibrotic (myo)fibroblast subsets and histological increased collagen deposition in the lamina muscularis in mice colon lacking HIF-1α in NKp46+ cells.**

**(A)** UMAP plots of mesenchymal cells zoom in with different fibroblast and smooth muscle cell subsets. **(B)** Dot plot of profibrotic gene and myofibroblast marker expression for each subset. **(C)** Top: the abundance of fibroblasts and smooth muscle cells in % of mesenchymal cells of HIF-1α KO (right) and WT (left). Bottom: the relative abundance of the different subsets of fibroblasts and smooth muscle cells. **(D)** GO terms and associated genes for subset Fibro 2. **(E)** RNA velocity analysis of

# Discussion

In summary, our data suggest that HIF-1–mediated ILC1 activation protects against excessive inflammation and the development of fibrosis during chronic intestinal damage. In our previous report (Krzywinska et al, 2022), loss of HIF-1α in NKp46⁺ leads to less severe acute colitis and weight loss after a single exposure to a high (3%) dose of DSS, owing to a reduced frequency of proinflammatory ILC1s and the relative abundance of protective NKp46⁺ ILC3s (Krzywinska et al, 2022). In contrast, in the current study, HIF-1α KO mice do not show a significantly reduced disease activity index after the first DSS exposure in the chronic colitis model (Fig 1A). However, it is important to point out that the chronic colitis model with repetitive exposure is carried out with 2% DSS, a dose that in our hands does not induce massive acute weight loss (data not shown) but allows us to study colitis and fibrogenesis in a chronic setting over an extended time frame. Therefore, the impact of HIF-1α in NKp46⁺ on intestinal damage might not become apparent upon the first exposure to a more modest dose in the chronic DSS model. Surprisingly, in the chronic colitis setting, HIF-1α KO mice show enhanced disease activity and intestinal damage (Fig 1A and B), despite a reduction in proinflammatory ILC1s (Fig 1C). This seems counterintuitive, as the severity of colitis is usually linked to the amplitude of type 1 inflammation (Neurath, 2019). However, the relative depletion of tissue-resident ILC1s and the reduced mRNA expression of the ILC1 signature cytokine Ifng in the colon of HIF-1α KO mice (Fig 1C and D) are in turn associated with increased infiltration of proinflammatory neutrophils (Fig 1E). It is therefore tempting to speculate that suppression of ILC1s, as the first line of acute and tissue-resident type 1 inflammation, may ameliorate acute colitis, yet lead to an "overshooting" type 1 immune response by enhanced infiltration of immune cells from the circulation and aggravated disease in the setting of chronic colitis. Our observations are, however, in line with various studies, showing that genetic or pharmacologic HIF stabilization in gut epithelial cells is protective against mucosal damage (Tambuwala et al, 2010, 2015). Likewise, pharmacologic HIF activation has been shown to be antifibrotic in chronically inflamed tissues (Steiner et al, 2022).

ILCs are highly plastic, and ILC subsets can interconvert in response to cytokines (Colonna, 2018). Notably, ILC3 plasticity has been demonstrated in mice and humans and NKp46⁺ ILC3s can convert into IFN-γ–producing NKp46⁺ ILC1s in response to IL-12 (Vonarbourg et al, 2010; Bernink et al, 2015; Forkel & Mjösberg, 2016). It is noteworthy that ILC3-to-ILC1 skewing in response to the cytokine IL-12 with increased levels of IFN-γ has been recognized as a pathogenic event during inflammatory bowel disease (Bernink et al, 2015; Forkel & Mjösberg, 2016). We have previously demonstrated in the context of acute colitis that loss of HIF-1α in NKp46⁺ cells prevents ILC3-to-ILC1 conversion, and confers protection against acute intestinal damage (Krzywinska et al, 2022). The ILC phenotype in the gut depends on exogenous cytokines and ILC-intrinsic RORγt/T-bet gradients, and we have demonstrated that the hypoxic response in NKp46⁺ cells drives acute phenotypic ILC changes in a HIF-1α–dependent manner. In the context of acute intestinal damage, HIF-1α contributes to T-bet expression and the ILC1 state, whereas loss of HIF-1α favors an increased RORγt/T-bet gradient and an ILC3 phenotype by direct transcriptional control of T-bet by HIF-1α (Krzywinska et al, 2022). Although we did not investigate ILC plasticity in detail in this study, the persistent relative depletion of ILC1s in HIF-1α KO mice after chronic DSS exposure suggests that the absence of HIF-1α in NKp46⁺ cells results in a sustained shift in the ILC1/ILC3 ratio. Yet, in contrast to acute DSS exposure, where a reduced ILC1/ILC3 ratio in HIF-1α KO mice is protective, in the context of chronic colitis, loss of HIF-1α in NKp46⁺ cells and prolonged reduction in ILC1s are associated with intestinal fibrosis.

In addition, we observe the abundance of putative fibrogenic mesenchymal subsets and intestinal fibrosis upon loss of HIF-1α in NKp46⁺ cells (Fig 3). Although the link between inflammation and intestinal fibrosis is less straightforward, as anti-inflammatory therapies ameliorate inflammation but do not appear to treat the fibrotic component (Rieder et, 2017), certain immune cell subsets are associated with the development of fibrosis; namely, macrophages can play an important role during fibrogenesis, owing to their plasticity and the ability to adopt different phenotypes depending on the microenvironmental challenges (Wynn & Vannella, 2016). For instance, the appearance and abundance of M2- versus M1-polarized macrophages have been typically associated with repair processes and scar formation (Sica & Mantovani, 2012). However, with regard to fibrogenic action, it has been shown that classifying macrophage populations by differential Ly6C expression can be more accurate than the M1/M2 classification to discern profibrotic macrophage phenotypes in vivo and in a tissue-dependent context (Ramachandran et al, 2012). Consistently, we observe an increase in Ly6Cʰⁱ macrophages upon HIF-1α deficiency in NKp46⁺ cells, whereas the abundance of M1/M2 macrophages is comparable across genotypes (Fig 1E and F). It is noteworthy that despite increased intestinal fibrosis in HIF-1α KO mice, we did not detect massive expansion of the mesenchymal cell compartment. Instead, we observe a shift from smooth muscle cells toward the fibroblasts within the mesenchymal cell compartment and trajectory analysis indeed suggests that smooth muscle cells increasingly acquire a fibroblast phenotype in the colon of HIF-1α KO mice (Fig 3). Although this would need to be formally demonstrated, for example, by lineage tracing, this hypothesis is in line with previous reports on the plasticity of mesenchymal cells in the context of inflammatory bowel disease (Li & Kuemmerle, 2014). Moreover, the fibrosis in the colon of HIF-1α KO mice dominantly occurs in the tunica muscularis and it is tempting to speculate that loss of smooth muscle cells at the expense of fibroblasts within the muscular layer might also contribute to reduced gut motility in the context of chronic colitis and intestinal fibrosis. HIF-1α deficiency in NKp46⁺ cells results in enhanced signaling along the BMP and the integrin pathway (Fig S3). Of note, both

---

mesenchymal cell comparison between HIF-1α KO (right) and WT (left). **(F)** Representative images and analysis of PDGFR-a+ area in the submucosa of WT and HIF-1α KO colon sections. **(G)** Analysis of average fibrotic thickness of WT and HIF-1α KO colon sections. Pooled data of two experiments; n KO = 5, n WT = 5; data are mean values ± SD; *P < 0.05, two-tailed t test. **(H)** Left: representative images of Sirius Red/Fast Green–stained lamina muscularis sections and enhanced collagen signal. Right: analysis of % Sirius Red–positive area in the lamina muscularis of the colon of HIF-1α KO and WT mice at the endpoint. Pooled data of three experiments; n WT ≥ 11, n KO ≥ 7; data are mean values ± SD; *P < 0.05 and ***P < 0.001, two-tailed t test.

pathways and their mutual interaction (Ashe, 2016) represent major profibrotic signaling hubs that drive fibroblast expansion across multiple organs (Henderson & Sheppard, 2013; Henderson et al, 2020). In line with this, the colon of HIF-1α KO mice is characterized by the abundance of fibroblasts with an enhanced expression of a profibrotic gene signature, including *Col1a1, Col1a2, Col3a1, Adamts3, Adamts9, Fn1, Postn* (Figs 3 and S3), and increased collagen deposition (Fig 3).

Taken together, our results identify HIF-1α in NKp46[+] cells as a double-edged sword and a critical nexus that orchestrates type 1 inflammation during acute colitis versus profibrotic responses in the context of chronic colitis. In this conceptual framework, adequate type 1 inflammation driven by HIF-1α in tissue-resident ILC1s is detrimental during acute colitis (Krzywinska et al, 2022). Yet, in the context of chronic colitis, HIF-1α in ILC1s counteracts excessive recruitment of proinflammatory neutrophils and Ly6C[hi] macrophages during chronic colitis, whereas loss of HIF-1α in NKp46+ ILCs results in increased intestinal fibrosis.

# Materials and Methods

### Animal models

The specific loss of HIF-1α in NKp46-expressing cells was accomplished by breeding mice with the HIF-1α gene surrounded by loxP sites with mice carrying the Cre recombinase enzyme under the control of the NKp46 promoter. Mice carrying the Cre recombinase are termed as HIF-1α KO, WT mice as WT. 8- to 12-wk-old mice were treated with 2% DSS in autoclaved and distilled drinking water. The treatment lasted for 3 d, followed by 4 d of the recovery phase with regular drinking water. This cycle was repeated four times. Weight and test for occult fecal blood have been recorded every 2nd d. All animal experiments were approved by the local animal ethics committee (Kantonales Veterinäramt Zürich in Switzerland) and performed according to local guidelines and the animal protection laws.

### Isolation of lamina propria cells

Colon without cecum and rectum of 8- to 12-wk-old mice were flushed with PBS, and adipose visceral tissue was removed. The colon was sliced longitudinally, and the luminal side was exposed. Feces were now removed by flushing with PBS. Colon pieces were stored in HBSS containing 2% FBS and 10 mM Hepes at 4°C until beginning with further dissociation. Epithelial cells were removed by rocking for 20 min at 37°C at 1.43*g* (80 rpm) in 5 mM dithiothreitol in HBSS containing 10 mM Hepes and 2% FBS. After incubation, colon pieces were washed with HBSS containing 10 mM Hepes, 2% FBS, and 5 mM EDTA. Samples were incubated for 15 min at 37°C with shaking at 1.43*g* (80 rpm). EDTA washing step was repeated twice. After the last washing step, colons were cut into small pieces using blunt-ended scissors and incubated in HBSS containing 100 μg/ml Liberase and 30 μg/ml DNase in 37°C with shaking at 7.25*g* (180 rpm) for 30 min. After incubation, tissues were further dissociated in gentleMACS Dissociator running the protocol "intestine." After dissociation, cells were filtered through a 100-μm cell strainer and centrifuged (O'Leary et al, 2021). The cell pellets were resuspended in PBS containing 2% FBS and 2 mM EDTA (FACS buffer). All centrifugations were performed using the Thermo Fisher Scientific TX-1000 rotor (75003017; Thermo Fisher Scientific).

### Flow cytometry

100 μl of lamina propria single-cell suspension was transferred to a 96-well plate. Cells were stained for 10 min at 4°C with CD16/32 in FACS buffer to prevent unspecific binding. Surface stainings, including Live/Dead marker, for ILCs and myeloid cells (Table 1) were done in FACS buffer for 20 min at 4°C. After surface staining, transcription factor and intracellular stainings were achieved using the Foxp3/transcription factor staining kit according to the manufacturer's instructions. The acquisition was performed on a BD FACSymphony 5L. Analysis was performed with FlowJo.

**Table 1.  Materials.**

| Reagent | Source or supplier | Catalog number |
|---|---|---|
| Animals | | |
| C57BL/6J | Charles River/JAXTM | N/A |
| Antibodies | | |
| FITC anti-mouse CD11c | BioLegend | 117306 |
| FITC anti-mouse CD19 | BioLegend | 152404 |
| FITC anti-mouse Ly6G | BioLegend | 127606 |
| FITC anti-mouse TER119 | BioLegend | 116206 |
| FITC anti-mouse TCR-β | BioLegend | 109206 |
| FITC anti-mouse TCR-γ/δ | BioLegend | 118106 |
| APC anti-mouse RORγt | eBioscience | 17-6981-82 |
| Alexa Fluor 700 anti-mouse CD45 | BioLegend | 103128 |
| APC-eFluor 780 anti-mouse CD90.2 (Thy-1.2) | eBioscience | 47-0902-82 |
| Brilliant Violet 421 anti-mouse CD127 | BioLegend | 135024 |

| Reagent | Source or supplier | Catalog number |
|---|---|---|
| LIVE/DEAD Fixable Aqua Dead Cell Stain Kit | Invitrogen | L34957 |
| Brilliant Violet 785 anti-mouse T-bet | BioLegend | 644835 |
| BUV395 anti-mouse NK-1.1 | BD Biosciences | 564144 |
| BUV737 anti-mouse CD49b | BD Biosciences | 741752 |
| BUV805 anti-mouse CD49a | BD Biosciences | 741976 |
| PE-eFluor 610 anti-mouse EOMES | eBioscience | 61-4875-82 |
| PE/Cyanine7 anti-mouse NKp46 | BioLegend | 137618 |
| Alexa Fluor 488 anti-mouse CD8a | BioLegend | 100723 |
| APC anti-mouse Ly6C | eBioscience | 17-5932-82 |
| BUV395 anti-mouse CD4 | BD Biosciences | 565974 |
| V450 anti-mouse Ly6G | BD Biosciences | 560603 |
| PE-Cy 7 anti-mouse CD11b | BD Biosciences | 552850 |
| PE anti-mouse Siglec-F | BioLegend | 155505 |
| PE/Cyanine5 anti-mouse TCRβ | BioLegend | 109210 |
| PE anti-mouse F4/80 | BioLegend | 123110 |
| PerCP-eFluor 710 anti-mouse iNOS | eBioscience | 46-5920-82 |
| FITC anti-mouse CD206 | BioLegend | 141703 |
| Purified anti-mouse CD16/32 | BioLegend | 101302 |
| Chemicals, enzymes, and other reagents | | |
| PBS | Thermo Fisher Scientific | 10010015 |
| RPMI | Thermo Fisher Scientific | 21870-076 |
| FBS | Thermo Fisher Scientific | 10500064 |
| Hepes | Thermo Fisher Scientific | 15630080 |
| Dithiothreitol | Thermo Fisher Scientific | R0861 |
| EDTA | Thermo Fisher Scientific | 15575020 |
| HBSS with Ca/Mg | Thermo Fisher Scientific | 24020117 |
| HBSS without Ca/Mg | Thermo Fisher Scientific | 14170088 |
| DNase | Roche | 10104159001 |
| Liberase | Roche | 5401127001 |
| Sirius Red | Sigma-Aldrich | 365548 |
| Fast Green | Merck | 1.04022.0025 |
| DPX mounting medium | Sigma-Aldrich | 06522 |
| Dextran sulfate sodium salt, MW 40,000 | Thermo Fisher Scientific | J63606 |
| Foxp3/Transcription Factor Staining Buffer Set | eBioscience | 00-5523-00 |
| RNeasy Mini Kit | QIAGEN | 74106 |
| High-Capacity cDNA Reverse Transcription Kit | Applied Biosystems | 4368814 |
| Software | | |
| GraphPad Prism 10 | https://www.graphpad.com/ | |
| ImageJ | https://imagej.net/ | |
| FlowJo | https://www.flowjo.com | |
| Other | | |
| hemoCARE Guajak-Test | CARE diagnostica | 005031 |
| GentleMACS Dissociator | Miltenyi Biotec | |
| LightCycler 96 | Roche | |

## Histology

Cleaned colon pieces were fixed for 24 h in 4% PFA at 4°C. After fixation, samples were embedded in paraffin and cut in slides with 5 µm thickness. Slides were deparaffinized using an automatic deparaffinator (Pathisto A-2). To investigate the degree of fibrosis, samples were stained with Sirius Red/Fast Green and Mayer's hematoxylin and eosin staining. Images were acquired using a Zeiss Axioscan slide scanner. Images were quantified using ImageJ v.1.54f software. For the histological degree of inflammation and damage, 4 features were determined: severity of inflammation (score 0–3), extent of inflammation (score 0–3), crypt damage (score 0–4), and percentage involvement (score 0–4); the first three parameters were multiplied by their percentage involvement (Tambuwala et al, 2010).

## Immunofluorescence

FFPE samples were deparaffinized as described above. After deparaffinization, antigen retrieval was performed in citrate buffer (C9999; Sigma-Aldrich) using a heat-induced method in Microwave Histoprocessor Histos Pro (Milestone). After cooling down, samples were washed with PBS. Blocking was achieved using blocking buffer containing 2% goat serum in PBS containing 0.05% Tween-20 for 1 h at RT. After blocking, slides were washed and mounted with primary antibody at 4°C overnight. The primary antibodies used were as follows: anti-LGR5 (1:100, PA5-23000; Invitrogen), anti-MUC2 (1:100, PA5-21329; Invitrogen), anti-PDGFRa (1:50, ab203491; Abcam). After primary antibody incubation, cells were incubated for 1 h at RT with secondary antibody (1:500, A11034; Invitrogen) in blocking buffer. For double staining, slides were stained with anti-alpha-smooth muscle actin (1:1,000, C6198; Sigma-Aldrich) at RT for 3 h. After the second primary antibody incubation, slides were incubated with DAPI for nuclear staining (1915865; Thermo Fisher Scientific) at 1 µg/ml in PBS for 10 min. After nuclear staining, slides were incubated with 0.1% Sudan black in 70% ethanol and mounted with a DPX mounting medium. Pictures were taken by a DMI 6000B microscope (Leica) and analyzed and quantified with ImageJ v.1.54f.

## qRT-PCR

Frozen parts of distal colon were homogenized in RLT buffer (QIAGEN). RNA was extracted using RNeasy Mini Kit (QIAGEN) following the manufacturer's instructions. Reverse transcription was achieved using the High-Capacity cDNA Reverse Transcription Kit (Applied Biosystems). For qRT-PCR, 10 ng cDNA with SYBR Green I Master Mix (Roche) was used. PCR conditions were as follows: 95°C for 10 min followed by 45 cycles of 95°C for 15 s and 60°C for 1 min. Data were normalized to 16s (primers: *16s*: forward primer, 5′-AGA TGA TCG AGC CGC GC-3′; reverse primer, 5′-GCT ACC AGG GCC TTT GAG ATG GA-3′; *IFN-γ*: forward primer, 5′-TCA AGT GGC ATA GAT GTG GAA GAA-3′; reverse primer, 5′-TGG CTC TGC AGG ATT TTC ATG-3′).

## Single-nucleus RNA-seq and data analysis

To isolate single nuclei from mouse colon, Sigma-Aldrich's Nuclei EZ Prep Kit (Cat #NUC-101; Sigma-Aldrich) was used. Briefly, snap-frozen mouse colon samples were firstly cut into 2-mm pieces,

homogenized using gentleMACS Dissociator (m_intestine_01 mode 2x, C Tubes) in 4 ml of ice-cold EZ Prep buffer (supplemented with 0.2 U/µl RNase inhibitor, #2313A; Clontech), and incubated on ice for 5 min. Subsequently, nuclei were filtered through 20-µm Pre-Separation Filter (130-101-812; Miltenyi Biotec) and centrifuged at 500*g* for 5 min at 4°C. After centrifugation, the nuclei were washed in 4 ml suspension buffer (1× PBS supplemented with 0.01% BSA and 0.2 U/µl RNase inhibitor [Cat #2313A; Clontech]). Isolated nuclei were resuspended in 1 ml suspension buffer, filtered through 20-µm Pre-Separation Filter (130-101-812; Miltenyi Biotec), and counted. A final concentration of 1,000 nuclei per µl was used for 10x Genomics Chromium loading. Considering a 10–20% loss, we pipetted 9.9 µl single-nucleus suspension (concentration of ~1,000 nuclei/µl, ~9,900 nuclei), targeting the recovery of ~6,000 nuclei. A single-nucleus suspension targeting 6,000 cells per sample was loaded into 10x Genomics Chromium Single Cell Controller and barcoded with unique molecular identifiers. Single-nucleus RNA-seq libraries were obtained following the 10x Genomics recommended protocol, using the reagents included in the Chromium Single Cell 3′v3.1 Reagent Kit. Libraries were sequenced on the NovaSeq 6000 (Illumina) instrument, aiming at 50 k mean reads per nuclei.

The 10x Genomics scRNA-seq data were processed using cellranger-7.0.1 with the mouse refdata-gex-mm10-2020-A. Based on the filtered gene–cell count matrix by CellRanger's default cell calling algorithm, we performed the Seurat workflow. To exclude low-quality cells and doublets, cells with less than 400 or more than 5,000 detected genes, and the percentage of mitochondrial gene counts higher than 10% were filtered out. Thirty principal components were chosen in the dimensionality reduction step. WT and KO samples were integrated using Seurat (version 4.3.0). A resolution of 1 (Seurat_clusters) in the unsupervised clustering resulted in 20 populations (data not shown). Ten major cell populations in colon were annotated by canonical marker genes of endothelial cells (*Pecam1*, *Kdr*, etc..), enteric neurons (*Ank2*, *Kcnip4*, etc...), enterocytes_Abs (*Caecam20*, *Plac8*, etc...), enterocytes_MP (*Cftr*, *Htr4*, etc...), enteroendocrine cells (*Kcnb2*, *Rimbp2*, etc...), goblet cells (*Muc2*, *Bcas1*, etc...), immune cells (*Ptprc*, *Ripor2*, etc...), intestinal stem cells (*Cenpe*, *Knl1*, etc...), lymphatic ECs (*Reln*, *Ntn1*, etc...), and mesenchymal cells (*Dmd*, *Cald1*, etc...). Mesenchymal nuclei were subset from total nuclei and further clustered using 30 principal components in the dimensionality reduction step and a supervised clustering step based on the expression level of myofibroblast or profibrotic smooth muscle cell marker list (*Tnc*, *Actg2*, *Acta2*, *Ptn*, *Des*, *Col5a1*, *Col4a1*, *Col3a1*, *Col1a1*, *Lamb1*, *Adam12*, *Eln*, *Fn1*, *Svep1*, *Postn*) (Bagalad et al, 2017). To determine the percentage of occupancy of different subclusters, we first calculated the proportions of each subcluster within each group (WT and KO), respectively; then, occupancy from WT and KO for each subcluster was calculated based on the normalized proportions. The top 10 markers of major cell populations in colon were identified by the FindAllMarkers function in Seurat (version 4.3.0) and displayed as a heatmap by the ComplexHeatmap function (version 2.12.1). Gene ontology analysis was performed based on the EnrichGO function in the clusterProfiler package (version 3.0.4). RNA velocity was estimated by the velocity python module (La Manno et al, 2018). Briefly, to generate loom files, spliced and unspliced reads were annotated by velocyto.py with CellRanger (version 7.0.1) output

files. The loom files were then loaded to Jupyter Notebook (Python version 3.9.13). The calculation of RNA velocity values for each cell and embedding RNA velocity vector to UMAP were done by following the scvelo python pipeline. CellChat analysis of the communications between different cells in colon was performed by the CellChat (Jin et al, 2021) package (version 1.6.1) in R program. Briefly, the interactions were identified and quantified based on the differentially overexpressed ligands and receptors for each cell population using "identifyOverExpressedGenes" and "identifyOverExpressedInteractions" functions. Cell populations less than 10 cells were filtered out using the "filterCommunication" function. "netVisual_diffInteraction" function was used to show the differences in the number of intercellular communication (number of interactions). Next, to compare the strength of different pathways between WT and KO, the function "rankNet" was employed. The comparison of the signaling heatmap between WT and KO was performed by the function "netAnalysis_signalingRole_heatmap." Finally, the communication probabilities of ligand–receptor pairs regulated by mesenchymal cells in other populations were compared in the function "netVisual_bubble."

## Statistical analysis

For statistical analysis, GraphPad Prism 10 was used. Statistical significance was calculated using the unpaired $t$ test. For qRT-PCR results, an outlier test (ROUT, Q = 1%) was performed. Sample numbers (n) are displayed in the figure legends. Statistical significance is displayed as $*P < 0.05$, $**P < 0.01$, and $***P < 0.001$.

# Data and Code Availability

The single-nucleus RNA-seq data of mouse colon reported in this study are deposited at the Gene Expression Omnibus (GEO) repository under the accession number GSE252602. All data are available in the main text or the supplementary materials, and raw data files that support the findings of this study are available from the corresponding author upon reasonable request.

# Supplementary Information

# Acknowledgements

We appreciate the support of the Swiss National Science Foundation (310030_179235), the Swiss Cancer League (KFS-4398-02-2018 and KFS-5402-08-2021), and the Swiss National Centre for Competence in Research "Kidney.CH" (N-403-06-26 HCP project grant) to C Stockmann. M Sobecki was supported by the internal postdoc funding program of the University of Zurich (Forschungskredit UZH Postdoc 2019). We thank the Laboratory Animal Services Center (LASC) for animal care and the UZH Irchel Flow Cytometry Core Facility. We acknowledge the support from the SKINTEGRITY.CH collaborative research program.

## Author Contributions

E Nelius: conceptualization, data curation, formal analysis, investigation, visualization, methodology, and writing—original draft, review, and editing.
Z Fan: conceptualization, data curation, formal analysis, investigation, methodology, and writing—original draft.
M Sobecki: conceptualization and methodology.
E Krzywinska: conceptualization and methodology.
S Nagarajan: conceptualization, formal analysis, and investigation.
I Ferapontova: conceptualization, investigation, and methodology.
D Gotthardt: conceptualization.
N Takeda: conceptualization.
V Sexl: conceptualization.
C Stockmann: conceptualization, resources, supervision, funding acquisition, methodology, project administration, and writing—original draft, review, and editing.

## Conflict of Interest Statement

The authors declare that they have no conflict of interest.

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
