## [Reviewer comments · Life Science Alliance]

Life Science Alliance

The transcription factor HIF-1 α in NKp46+ ILCs limits chronic intestinal inflammation and fibrosis

Eric Nelius, Zheng Fan, Michal Sobocki, Ewelina Krzywinska, Shunmugam Nagarajan, Irina Ferapontova, Dagmar Gotthardt, Norihiko Takeda, Veronika Sexl, and Christian Stockmann

DOI: <https://doi.org/10.26508/lsa.202402593>

Corresponding author(s): Christian Stockmann, University of Zurich; Christian Stockmann, University of Zurich; and Eric Nelius, University of Zurich

Review Timeline:

Submission Date:	2024-01-13
Editorial Decision:	2024-02-22
Revision Received:	2024-05-06
Editorial Decision:	2024-05-28
Revision Received:	2024-06-04
Accepted:	2024-06-05

Transaction Report:

February 22, 2024

Re: Life Science Alliance manuscript #LSA-2024-02593-T

Prof. Christian Stockmann
University of Zurich
SWITZERLAND

Dear Dr. Stockmann,

Thank you for submitting your manuscript entitled "Expression of the transcription factor HIF-1 α in NKp46+ ILCs limits intestinal fibrosis" to Life Science Alliance. The manuscript was assessed by expert reviewers, whose comments are appended to this letter. We invite you to submit a revised manuscript addressing the Reviewer comments.

Thank you for this interesting contribution to Life Science Alliance. We are looking forward to receiving your revised manuscript.

Sincerely,

B. MANUSCRIPT ORGANIZATION AND FORMATTING:

Reviewer #1 (Comments to the Authors (Required)):

The authors studied the impact of HIF-1 in ILCs in a mouse model of chronic/repetitive colitis. To this aim, they used a NKp46-promotor driven Cre to specifically knock out the HIF-1a allele and treated mice with a moderate dose of DSS to mimic the repetitive inflammation as seen in chronic colitis.

In confirmation of their previous work, HIF-1-mediated ILC1 activation was detrimental in the acute phase of colitis but appeared protective in the chronic phase of the disease. Increased numbers of neutrophils and a profibrotic signature in HIF-1a deficient mice convincingly demonstrated this.

As demonstrated in Fig 1a the authors provide a valid experimental model with the unexpected finding of a biphasic effect of HIF-1-deficient ILC1. The later, chronic phase is then corroborated by data on the immune cell subtype in Fig1 c to f. Fig 2 extends these findings by scRNA sequencing data to provide evidence for a substantially compromised signaling in the regulatory network between epithelial and mesenchymal cells in the colon. This leads to (Fig 3) the further central finding in this model that expansion of profibrotic fibroblasts contributes to increased fibrosis as demonstrated by enhanced Sirius Red staining in Fig 3f.

In general, this is a very nicely worked out hypothesis driven study with an unexpected biphasic role for HIF-1 in ILC1 during chronic colitis. The findings are new and of importance, particularly in view of drugs that have entered the market to increase HIF-1 protein but also those which are aiming at inhibiting HIF-1. Any disease with a biphasic character of HIF-1 action will need specific attention when considering any kind of the above-mentioned treatments.

A few things I would like the authors to address:

1. Is there any evidence that the barrier function of the colon epithelium is negatively affected in the chronic phase in HIF-1a-Kos?
2. Bacteria play a significant role during colitis - did the authors see differences in submucosal bacterial invasion?
3. Where there any compensatory changes in HIF-2 in their ILCs, e.g. by altered PHD expression?

Reviewer #2 (Comments to the Authors (Required)):

Nelius et.al utilize the chemically induced DSS model to investigate the ILC-specific role of HIF-1 α in chronic intestinal inflammation and fibrosis. Previous work by the authors has demonstrated the influence of HIF-1 α on a pathogenic ILC phenotype in acute DSS-induced colitis. In this study, the authors investigate the ILC-specific function of HIF-1 α within the chronic context of the DSS model, suggesting a protective role for this transcription factor in chronic colitis and fibrosis. A few major points should be addressed in support of the conclusions reached by the study:

1. The authors attribute the deletion of HIF-1 α using the NCR1-cre to HIF-1 α depletion in ILCs, assuming specificity. Given that NCR1 also targets NK cells (Curio et al., 2022), the authors are requested either to document the specificity of the Cre line in the colon or attribute their findings to all NCR1+ cells.
2. While the inflammation data are convincing, the fibrosis data (Figure 3) are poor. The lineage trajectory and fibroblast composition data from single-nuclei analysis lack statistical power due to the limited number of cells and comparisons based on single mice. Additional validation, such as flow cytometry or imaging, is necessary to confirm myofibroblast expansion.
3. The authors choose the non-classical DSS model instead of the widely used chronic DSS colitis model (3 cycles of 2.5% DSS, Jasso et al., 2022), which exhibits fibrosis in the submucosal layer (increased thickness, myofibroblast expansion, collagen deposition). Consequently, they are requested to assess the extent of fibrosis development in the non-classical DSS model and provide representative data of the intestinal fibrotic pathology.

Additional points:

1. Figure 1D's y-axis values are unclear. The authors should clarify whether the expression of the *Ifn γ* gene is presented relative to a control and consider using fold change instead. Additionally, explain the rationale behind excluding outliers using the ROUT method.
2. The authors state that 'The deletion of HIF-1 α in NKp46+ cells leads to depletion of ILC1s from the colon, similar to the acute colitis setting'. Given that the flow cytometric analysis of ILCs in Figure 1 shows differences between the WT and HIF-1 α KO DSS-challenged mice in terms of percentages, but not absolute numbers, it would be more accurate to replace the word 'depletion' with 'relative depletion' or with the ratio of NKp46+ ILC3/ILC1 cells. By including such a graph, the results will correspond better to the data published in the paper by Krzywinska et al., 2022.
3. A graph showing the absolute total number of CD45+ cells between the two genotypes upon DSS challenge needs to be included in Figure A. This would be informative for the single nuclei-RNA sequencing results, to exclude the possibility that the

observed differences in the cluster abundances result from cell type dilution due to the overrepresentation of immune cells in the sample.

4. Including an extra visualization of the cluster abundances complementary to the UMAP in Figure 2 (e.g. bar plot) will help to make the differences distinguishable to the reader's eye.

5. It is important to provide a validation of the observed differences in frequency of mucus-producing enterocytes and intestinal stem cells between the genotypes upon chronic DSS treatment (e.g. through flow cytometry or imaging), as it will confer additional value to the snRNA-seq data presented in the manuscript.

6. With respect to the results of Figure 1, the authors should consider discussing the possibility of intra-subset plasticity between ILC3s and ILC1s.

7. The authors state 'Yet, in the context of chronic colitis HIF-1 α in ILC1s' counteracts excessive recruitment of proinflammatory neutrophils and profibrotic macrophages during chronic colitis'. Given that the pro-fibrotic nature of the Ly6high macrophages is not supported by any data in the current manuscript, the authors should consider avoiding this overstatement in their conclusions.

8. Given that the individual values are not plotted in Figure 1A, the authors should consider expressing the data as mean values {plus minus} SD instead of SEM.

9. In Figure 1F, the Ly6Clow gating includes both the Ly6C negative and Ly6C low-expressing macrophages. Hence, it would be more accurate to label the population as Ly6Cneg/low.

10. In Figures 2A and 3A, the authors need to increase the font size so that the cluster labels are distinguishable.

11. Since Figure 2D is a quite busy plot, separating the ligand-receptor interactions into super-categories would render the graph more reader-friendly.

Reviewer #3 (Comments to the Authors (Required)):

In this manuscript, Nelius et al. have investigated the role of HIF-1 α in innate lymphoid cells (NKp46+). The authors demonstrate that HIF-1 α plays an important role in determining the intestinal innate lymphoid cell phenotype in the context of inflammatory bowel disease. Mice lacking HIF-1 α in ILCs have decreased ILC levels but increased neutrophils and macrophages and enhanced interaction mesenchymal cells with other compartments and decreased goblet cells and stem cells. The mice experienced increased BMP signaling and fibrosis. The authors conclude that ILC HIF-1 α is detrimental in acute inflammatory disease of the intestine but beneficial in chronic disease. The mechanisms by which HIF-1 α regulates immune cell function is an area of importance. While somewhat descriptive, the current manuscript adds to our understanding of the role of HIF in immune cell function, particularly in the context of intestinal inflammation.

Points for the authors

1) Can the authors be clear about the source of the mice and the nature of the confirmation that the knockout is effective and specific for NKp46-expressing ILCs.

2) The authors express surprise in the introduction that HIF-KO in ILCs is detrimental in chronic colitis, however many studies are consistent with this (i.e. multiple studies have shown that HIF activation with PHD inhibitors or VHL deletion. E.g. PMID: 18166353) is highly protective in colitis. The authors current data is actually consistent with this large volume of published data, some of which should be cited. Similarly, pharmacologic HIF activation has been shown to be anti-fibrotic in the chronically inflamed tissue (PMID: 27789456).

3) Do the authors have any information on the impact of HIF-1 α KO in ILCs on intestinal barrier function in these mice? This is important as it is the loss of barrier that is the key innate immune dysregulation in IBD.

4) Please include reference to the powering of experiments and the n numbers of mice used in the Statistics section.

From: lsa@msubmit.net <lsa@msubmit.net>
Sent: Thursday, February 22, 2024 5:48 PM
To: Christian Stockmann <christian.stockmann@anatomy.uzh.ch>
Subject: Life Science Alliance Manuscript - Editorial Decision LSA-2024-02593-T

February 22, 2024

Re: Life Science Alliance manuscript #LSA-2024-02593-T

Prof. Christian Stockmann
University of Zurich
SWITZERLAND

Dear Dr. Stockmann,

Thank you for submitting your manuscript entitled "Expression of the transcription factor HIF-1 α in NKp46+ ILCs limits intestinal fibrosis" to Life Science Alliance. The manuscript was assessed by expert reviewers, whose comments are appended to this letter. We invite you to submit a revised manuscript addressing the Reviewer comments.

Thank you for this interesting contribution to Life Science Alliance. We are looking forward to receiving your revised manuscript.

Sincerely,

Eric Sawey, PhD
Executive Editor

Life Science Alliance
<http://www.lsjournal.org>

- A letter addressing the reviewers' comments point by point.
- An editable version of the final text (.DOC or .DOCX) is needed for copyediting (no PDFs).
- High-resolution figure, supplementary figure and video files uploaded as individual files: See our detailed guidelines for preparing your production-ready images, <https://www.life-science-alliance.org/authors>
- Summary blurb (enter in submission system): A short text summarizing in a single sentence the study (max. 200 characters including spaces). This text is used in conjunction with the titles of papers, hence should be informative and complementary to the title and running title. It should describe the context and significance of the findings for a general readership; it should be written in the present tense and refer to the work in the third person. Author names should not be mentioned.
- By submitting a revision, you attest that you are aware of our payment policies found here: <https://www.life-science-alliance.org/copyright-license-fee>

B. MANUSCRIPT ORGANIZATION AND FORMATTING:

We would like to thank the editor for the exquisite editorial guidance and the opportunity to address the reviewer's concerns in a revised manuscript.

We very much appreciate the constructive and helpful comments from the reviewers, which definitely helped to improve the manuscript. We agree that additional efforts were required to support the manuscripts conclusion and after reading the reviewer's comments carefully, we performed the following key experiments to address the main criticism from the reviewers:

Quantitative assessment of epithelial barrier integrity and bacterial invasion by means of immunostaining.

Quantitative analysis of submucosal fibroblasts and mucosal thickness by immunostaining and mucosal thickness, respectively.

Quantitative assessment of intestinal stem cells and mucus-producing cells by means of immunostaining.

Reviewer #1 (Comments to the Authors (Required)):

The authors studied the impact of HIF-1 in ILCs in a mouse model of chronic/repetitive colitis. To this aim, they used a NKp46-promotor driven Cre to specifically knock out the HIF-1a allele and treated mice with a moderate dose of DSS to mimic the repetitive inflammation as seen in chronic colitis.

In confirmation of their previous work, HIF-1-mediated ILC1 activation was detrimental in the acute phase of colitis but appeared protective in the chronic phase of the disease. Increased numbers of neutrophils and a profibrotic signature in HIF-1a deficient mice convincingly demonstrated this.

As demonstrated in Fig 1a the authors provide a valid experimental model with the unexpected finding of a biphasic effect of HIF-1-deficient ILC1. The later, chronic phase is then corroborated by data on the immune cell subtype in Fig1 c to f. Fig 2 extends these finding by scRNA sequencing data to provide evidence for a substantially compromised signaling in the regulatory network between epithelial and mesenchymal cells in the colon. This leads to (Fig 3) the further central finding in this model that expansion of profibrotic fibroblasts contributes to increased fibrosis as demonstrated by enhanced Sirius Red staining in Fig 3f.

In general, this is a very nicely worked out hypothesis driven study with an unexpected biphasic role for HIF-1 in ILC1 during chronic colitis. The findings are new an of importance, particularly in view of drugs that have entered the market to increase HIF-1 protein but also those which are aiming at inhibiting HIF-1. Any disease with a biphasic character of HIF-1 action will need specific attention when considering any kind of the above-mentioned treatments.

Our Reply: We appreciate the overall positive perception of our manuscript as well as the very constructive and helpful comments.

A few things I would like the authors to address:

1. Is there any evidence that the barrier function of the colon epithelium is negatively affected in the chronic phase in HIF-1a-Kos?

Our

Reply:

We appreciate this question and performed staining for the tight junction protein Occludin in the epithelial cell layer on colon sections from WT and HIF1a KO mice after chronic DSS exposure at endpoint (day 28). As shown below, the quantification of the Occludin positive area and signal intensity in the mucosa did not reveal a statistically significant change in Occludin expression between the two genotypes:

Although, this does not represent a functional assay to assess epithelial barrier function, this suggests that at least at this time point, epithelial barrier integrity is not negatively affected after chronic DSS exposure by the loss of HIF1a in NKp46+ cells. This is further corroborated the assessment of mucosal bacterial invasion (just below). We feel that this data could contribute to the discussion section of the manuscript but would like to leave it to the reviewers and the editor, whether these results should be included in the manuscript.

2. Bacteria play a significant role during colitis - did the authors see differences in submucosal bacterial invasion?

Our Reply:

The reviewer raises an important point. We addressed this and performed Gram staining to detect Gram positive and negative bacteria on colon sections from WT and HIF1a KO mice after chronic DSS exposure at endpoint (day 28). As shown below, the quantification of bacteria in the mucosa did not reveal a statistically significant difference in bacterial invasion across genotypes. Of note, we were not able to detect any bacteria in the submucosa at this given time point after chronic DSS.

In summary, this suggests that the epithelial barrier integrity is not further compromised by the loss of HIF1a in NKp46+ cells. As outlined above, we feel that this data could contribute to the discussion section of the manuscript but would like to leave it to the reviewers and the editor, whether these results should be included in the manuscript.

3. Where there any compensatory changes in HIF-2 in their ILCs, e.g. by altered PHD expression?

Our Reply:

We appreciate this question. According to the IMMGEN database (<http://rstats.immgen.org/Skyline/skyline.html>), *Epas1* (the gene that encodes for HIF2a) is not expressed in NKp46+ ILC1s and NK cells and only at low level in NKp46+ ILC3s (see below). We have tried in previous projects to investigate the impact of HIF-2 in NKp46+ cells in the context of wound healing in the skin (Sobecki et al. 2021) and acute intestinal damage (Krzywinska et al. 2022). However, in contrast to loss of HIF1a, the deletion of *Epas1* in NKp46+ cells did not affect wound healing in the skin (Sobecki et al. 2021). When we initially tested the impact *Epas1* in NKp46+ on acute colitis after DSS exposure, the deletion of *Epas1* had no effect on acute colitis (see

below, unpublished data) and was, hence, not followed up further. Finally, we were not able to detect HIF2a in NKp46+ ILCs at the protein level. Therefore, we conclude that HIF-2 seems to be not expressed or only at only very low level and if at all plays only a minor role NKp46+ cell function.

Reviewer #2 (Comments to the Authors (Required)):

Nelius et.al utilize the chemically induced DSS model to investigate the ILC-specific role of HIF-1 α in chronic intestinal inflammation and fibrosis. Previous work by the authors has demonstrated the influence of HIF-1 α on a pathogenic ILC phenotype in acute DSS-induced colitis. In this study, the authors investigate the ILC-specific function of HIF-1 α within the chronic context of the DSS model, suggesting a protective role for this transcription factor in chronic colitis and fibrosis. A few major points should be addressed in support of the conclusions reached by the study:

Our Reply: We appreciate the overall positive perception of our manuscript as well as the very thoughtful and constructive comments.

1. The authors attribute the deletion of HIF-1 α using the NCR1-cre to HIF-1 α depletion in ILCs, assuming specificity. Given that NCR1 also targets NK cells (Curio et al., 2022), the authors are requested either to document the specificity of the Cre line in the colon or attribute their findings to all NCR1+ cells.

Our Reply:

We appreciate this comment and agree that the wording is not sufficiently accurate. The NCR1 (NKp46)-Cre is expressed in NKp46+ ILC3s and all ILC1s, including non-NK cell ILC1s and NK cells and we have indeed used the NCR1 (NKp46)-Cre in previous studies to delete HIF1a in NK cells (Krzywinska et al. 2017, Sobocki et al. 2021). We have therefore changed the wording in the manuscript as follows:

In order to test the role of HIF-1 α in NKp46+ gut ILCs in chronic injury in the large intestine, we used mice with a targeted deletion of HIF-1 α , via crosses of the loxP-flanked HIF-1 α allele to the *Ncr1* (NKp46) promoter-driven Cre recombinase, specific to NKp46-expressing ILC1s, including NK cells and ILC3s (*Ncr1*^{cre+} *Hif-1 α* ^{fl+/fl+} mice, termed HIF-1 α KO) (Eckelhart *et al.*, 2011; Krzywinska *et al.*, 2017) and wildtype (WT) littermates to a scheme of repetitive dextran sodium sulfate (DSS) exposure (Fig. 1a), which induces mucosal damage and chronic colitis (Okayasu *et al.*, 1990).

2. While the inflammation data are convincing, the fibrosis data (Figure 3) are poor. The lineage trajectory and fibroblast composition data from single-nuclei analysis lack statistical power due to the limited number of cells and comparisons based on single mice. Additional validation, such as flow cytometry or imaging, is necessary to confirm myofibroblast expansion.

Our Reply:

We appreciate this question and agree with the reviewer that additional validation of fibroblast expansion would strengthen the conclusion of the manuscript. Therefore, we have performed and quantified immunostainings for alpha-SMA and PDGFR-alpha on colon sections from WT and HIF1a KO mice after chronic DSS exposure at endpoint (day 28). As depicted below, the submucosa of HIF1a KO mice showed an increase in the PDGFR-alpha+ area.

Along with this, the thickness of the submucosal layer in colon of HIF1a KO mice is increased. Taken together, this further supports the conclusion that loss of HIF1a in NKp46+ cells leads to increased intestinal fibrosis upon chronic DSS exposure.

The results have been incorporated into the revised manuscript as Figure 3 F and G as follows:

This was further corroborated immunostainings for alpha-SMA and PDGFR-alpha on colon sections. As shown in Figure 3f the submucosa of HIF1a KO mice showed an increase in the PDGFR-alpha+ area along with increased thickness of the colon wall (Fig. 3g). Finally, Sirius red staining on colon tissue from WT and HIF-1 α KO mice at endpoint after chronic DSS exposure to detect the collagen distribution., the tunica muscularis of HIF-1 α KO mice shows increased collagen deposition (Fig. 3h).

3. The authors choose the non-classical DSS model instead of the widely used chronic DSS colitis model (3 cycles of 2.5% DSS, Jasso et al., 2022), which exhibits fibrosis in the submucosal layer (increased thickness, myofibroblast expansion, collagen deposition). Consequently, they are requested to assess the extent of fibrosis development in the non-classical DSS model and provide representative data of the intestinal fibrotic pathology.

Our Reply:

We appreciate this comment and would like to briefly explain, why we modified the protocol for chronic DSS exposure. In our hands, when using the chronic DSS colitis model (3 cycles of 2.5% DSS, Jasso et al., 2022), about 70% of the wildtype loose 15% of their initial bodyweight at day of the 1st DSS exposure. According to the swiss laws on animal experimentation, we have to terminate the experiment and euthanize the mice as soon as the mice loose 15% of their initial bodyweight, which precluded us from studying the effects of chronic DSS exposure and forced us to adapt the experimental protocol.

However, as demonstrated above (and in Figure 3 F, G and H) our adapted protocol also leads to fibroblasts expansion, increased mucosal thickness and collagen deposition.

Additional points:

1. Figure 1D's y-axis values are unclear. The authors should clarify whether the expression of the Ifny gene is presented relative to a control and consider using fold change instead. Additionally, explain the rationale behind excluding outliers using the ROUT method.

Our Reply:

We appreciate the reviewer's remark and have changed the graph in Figure 1D accordingly to "fold change" as shown below.

We have chosen the ROUT (Robust regression and Outlier removal) test to identify and remove outliers owing to several advantages over alternative methods that have been attributed to the ROUT test.

This method uses a robust regression approach which is less sensitive to outliers compared to traditional methods like linear regression. This robustness is important because it helps in accurately identifying outliers without being influenced by them. ROUT typically uses the False Discovery Rate (FDR) method to control the proportion of falsely identified outliers. This objective criterion helps in making consistent and replicable decisions about which points to exclude.

ROUT is considered to provide a good balance between data integrity and statistical accuracy. It excludes only those points that have a high likelihood of being genuine outliers, thus preserving the integrity of the dataset as much as possible while enhancing the accuracy of statistical analyses.

2. The authors state that 'The deletion of HIF-1 α in NKp46+ cells leads to depletion of ILC1s from the colon, similar to the acute colitis setting'. Given that the flow cytometric analysis of ILCs in Figure 1 shows differences between the WT and HIF-1 α KO DSS-challenged mice in terms of percentages, but not absolute numbers, it would be more accurate to replace the word 'depletion' with 'relative depletion' or with the ratio of NKp46+ ILC3/ILC1 cells. By including such a graph, the results will correspond better to the data published in the paper by Krzywinska et al., 2022.

Our Reply:

We agree with the reviewer and have changed the wording accordingly to "relative depletion" throughout the manuscript.

3. A graph showing the absolute total number of CD45+ cells between the two genotypes upon DSS challenge needs to be included in Figure A. This would be informative for the single nuclei-RNA sequencing results, to exclude the possibility that the observed differences in the cluster abundances result from cell type dilution due to the overrepresentation of immune cells in the sample.

Our Reply:

We appreciate this comment and have now included a graph showing the total number of CD45+ cells (see below) into Figure 1C of the revised manuscript and described as follows:

Of note, the total number of CD45+ cells, the frequencies of NKp46+-ILC3, NK cells as well as CD4+ and CD8+ T cells as assessed by flow cytometry were similar across genotypes (Fig. 1c and e, for gating strategy, please see Suppl. Fig. 1a and b).

4. Including an extra visualization of the cluster abundances complementary to the UMAP in Figure 2 (e.g. bar plot) will help to make the differences distinguishable to the reader's eye.

Our Reply:

We appreciate this suggestion and included the requested bar plot shown below into the revised manuscript as Figure 2B:

5. It is important to provide a validation of the observed differences in frequency of mucus-producing enterocytes and intestinal stem cells between the genotypes upon chronic DSS treatment (e.g. through flow cytometry or imaging), as it will confer additional value to the snRNA-seq data presented in the manuscript.

Our Reply:

We would like to thank the reviewer for raising this important point. In order to address this concern, we have quantified immunostainings for LGR5 (a marker for intestinal stem cells, Barker et al, 2007) as well as mucin 2 (MUC2, a marker for mucus-producing goblet cells and mucus-producing enterocytes, Pelaseyed et al, 2014) on colon sections from WT and HIF1a KO mice after chronic DSS exposure at endpoint (day 28).

As shown below, the colon of HIF1a KO mice exhibits a reduction of LGR5+ intestinal stem cells.

Moreover, in the colon of HIF1a KO mice, the area of MUC2+ cells was significantly reduced. MUC-2 is expressed by goblet cells as well as mucus-producing enterocytes and, therefore does not allow to distinguish the two cell types. However, given that the

number of goblet cells was not reduced according to the single-cell RNA sequencing analysis, this data indicates a reduction of mucus-producing cells, including enterocytes, in the colon of HIF1a KO mice.

The results have been included into Figure 2B of the revised manuscript and described as follows:

This observation is further supported by immunostainings for LGR5 (a marker for intestinal stem cells, (Barker et al, 2007) as well as mucin 2 (MUC2, a marker for mucus-producing goblet cells and mucus-producing enterocytes, Pelaseyed et al, 2014) on colon sections from WT and HIF1a KO mice after chronic DSS exposure at endpoint (day 28). As shown in Figure 2b, the colon of HIF1 α KO mice exhibits a reduction of LGR5+ intestinal stem cells as well as the area of MUC2+ cells. Of note, MUC-2 is expressed by goblet cells as well as mucus-producing enterocytes and, therefore does not allow to distinguish the two cell types. However, given that the number of goblet cells was not reduced according to the single-cell RNA sequencing analysis, this data indicates a reduction of mucus-producing cells, including enterocytes, in the colon of HIF1 α KO mice.

6. With respect to the results of Figure 1, the authors should consider discussing the possibility of intra-subset plasticity between ILC3s and ILC1s.

Our Reply:

We appreciate the reviewer's suggestions and have included the following paragraph into the discussion of the manuscript:

ILCs are highly plastic and ILC subsets can interconvert in response to cytokines (Colonna, 2018). Notably, ILC3 plasticity has been demonstrated in mice and humans and NKp46⁺ ILC3s can convert into IFN- γ -producing NKp46⁺ ILC1s in response to IL-12 (Vonarbourg et al., 2010; Bernink et al., 2015; Forkel and Mjösberg, 2016). Noteworthy, ILC3 to ILC1 skewing in response to the cytokine IL-12 with increased levels of IFN- γ has been recognized as a pathogenic event during inflammatory bowel disease (Forkel and Mjösberg, 2016; Bernink et al., 2015). We have previously demonstrated in the context of acute colitis that Loss of HIF-1 α in NKp46⁺ cells prevents ILC3-to-ILC1 conversion, and confers protection against acute intestinal damage (Krzywinska et al., 2022). The ILC phenotype in the gut depends on exogenous cytokines and ILC-intrinsic ROR γ t/T-bet gradients and we have demonstrated that the hypoxic response in NKp46⁺ cells drives acute phenotypic ILC changes in a HIF-1 α -dependent manner. In the context of acute intestinal damage, HIF-1 α contributes to T-bet expression and the ILC1 state, whereas loss of HIF-1 α favours an increased ROR γ t/T-bet gradient and an ILC3 phenotype by direct transcriptional control of T-bet by HIF-1 α (Krzywinska et al., 2022). Although, we did not investigate ILC plasticity in detail in this study, the persistent relative depletion of ILC1s in HIF-1 α KO mice after chronic DSS exposure suggests that absence of HIF-1 α in NKp46⁺ cells results in a sustained shift in the ILC1/ILC3 ratio. Yet, in contrast to acute DSS exposure, where a reduced ILC1/ILC3 ratio in HIF-1 α KO mice is protective, in the context chronic colitis, loss HIF-1 α in NKp46⁺ cells and prolonged reduction of ILC1s is associated with intestinal fibrosis.

7. The authors state 'Yet, in the context of chronic colitis HIF-1 α in ILC1s' counteracts excessive recruitment of proinflammatory neutrophils and profibrotic macrophages during chronic colitis'. Given that the pro-fibrotic nature of the Ly6high macrophages is not supported by any data in the current manuscript, the authors should consider avoiding this overstatement in their conclusions.

Our Reply:

We thank the reviewer for this suggestion and changed the wording in the revised manuscript as follows:

Yet, in the context of chronic colitis HIF-1 α in ILC1s counteracts excessive recruitment of proinflammatory neutrophils and Ly6C^{hi} macrophages during chronic colitis, whereas loss HIF-1 α in NKp46+ ILCs results in increased intestinal fibrosis.

8. Given that the individual values are not plotted in Figure 1A, the authors should consider expressing the data as mean values {plus minus} SD instead of SEM.

Our Reply:

We have changed the respective graphs in Figure 1A accordingly as shown below:

9. In Figure 1F, the Ly6C^{low} gating includes both the Ly6C negative and Ly6C low-expressing macrophages. Hence, it would be more accurate to label the population as Ly6C^{neg/low}.

Our Reply:

We appreciate this comment and would like to specify that all cells are Ly6C positive. Due to the smoothing function (to make the populations clearer for the readers eye) it appears like some cells are negative, nevertheless, all cells included here are Ly6C positive and are gated on a negative control.

10. In Figures 2A and 3A, the authors need to increase the font size so that the cluster labels are distinguishable.

Our Reply:

We have increased the font size in the respective figures, as requested by the reviewer.

11. Since Figure 2D is a quite busy plot, separating the ligand-receptor interactions into super-categories would render the graph more reader-friendly.

Our Reply:

We have rearranged the plot according to the reviewer's suggestion as shown below:

Reviewer #3 (Comments to the Authors (Required)):

In this manuscript, Nelius et al. have investigated the role of HIF-1alpha in innate lymphoid cells (NKp46+). The authors demonstrate that HIF-1alpha plays an important

role in determining the intestinal innate lymphoid cell phenotype in the context of inflammatory bowel disease. Mice lacking HIF-1 α in ILCs have decreased ILC levels but increased neutrophils and macrophages and enhanced interaction mesenchymal cells with other compartments and decreased goblet cells and stem cells. The mice experienced increased BMP signaling and fibrosis. The authors conclude that ILC HIF-1 α is detrimental in acute inflammatory disease of the intestine but beneficial in chronic disease. The mechanisms by which HIF-1 α regulates immune cell function is an area of importance. While somewhat descriptive, the current manuscript adds to our understanding of the role of HIF in immune cell function, particularly in the context of intestinal inflammation.

Our Reply:

We appreciate the overall positive perception of our manuscript as well as the very considerate and constructive comments.

Points for the authors

1) Can the authors be clear about the source of the mice and the nature of the confirmation that the knockout is effective and specific for NKp46-expressing ILCs.

Our Reply:

We appreciate this comment. The NKp46-cre that we use has been created, characterized and published by Eckelhardt et al. (Eckelhardt, E. et al. A novel Ncr1-Cre mouse reveals the essential role of STAT5 for NK-cell survival and development. *Blood* 117, 1565–1573 (2011)). We then created an in vivo, targeted deletion of HIF-1 α in NK cells, via crosses of the loxP-flanked HIF-1 α allele (Ryan, H. E. et al. Hypoxia-inducible factor-1 α is a positive factor in solid tumor growth. *Cancer Res.*, 4010–4015 (2000)) to the Ncr1 (NKp46) promoter-driven Cre recombinase (Eckelhardt et al., 2011) specific to NKp46-expressing innate lymphoid cells, including NK cells. Non-NK cell ILC1s and NKp46+ ILC3s (Narni-Mancinelli, E. et al. Fate mapping analysis of lymphoid cells expressing the NKp46 cell surface receptor. *Proc. Natl. Acad. Sci. USA* 108, 18324–18329 (2011); Eberl, G., Colonna, M., Di Santo, J. P. & McKenzie, A. N. Innate lymphoid cells: a new paradigm in immunology. *Science* 348, aaa6566 (2015)). This results in efficient deletion of HIF-1 α in isolated splenic NK cells, tumor-infiltrating NK cells (Krzywinska et al. 2017), skin wound-infiltrating NK cells (Sobecki et al. 2021) and intestinal NKp46+ ILCs at steady state conditions as well during acute intestinal damage (Krzywinska et al. 2017).

2) The authors express surprise in the introduction that HIF-KO in ILCs is detrimental in chronic colitis, however many studies are consistent with this (i.e. multiple studies have shown that HIF activation with PHD inhibitors or VHL deletion. E.g. PMID: 18166353) is highly protective in colitis. The authors current data is actually consistent with this large volume of published data, some of which should be cited. Similarly, pharmacologic HIF activation has been shown to be anti-fibrotic in the chronically inflamed tissue (PMID: 27789456).

Our Reply:

We would like to thank the reviewer for this comment. In order to address this, we have added the following paragraphs to the introduction and the discussion of the manuscript, respectively:

Multiple studies have shown that HIF activation in intestinal epithelial cells can protect from colitis (Cummins et al 2008, Tambuwala et al., 2010, 2015).

Our observation are, however, in line with various studies showing that genetic or pharmacologic HIF stabilization in gut epithelial cells is protective against mucosal damage (Cummins et al 2008, Tambuwala et al., 2010, 2015). Likewise, pharmacologic HIF activation has been shown to be anti-fibrotic in chronically inflamed tissues (Steiner et al 2022).

3) Do the authors have any information on the impact of HIF-1alpha KO in ILCs on intestinal barrier function in these mice? This is important as it is the loss of barrier that is the key innate immune dysregulation in IBD.

Our Reply:

We appreciate this question and performed staining for the tight junction protein Occludin in the epithelial cell layer on colon sections from WT and HIF1a KO mice after chronic DSS exposure at endpoint (day 28). As shown below, the quantification of the Occludin positive area and signal intensity in the mucosa did not reveal a statistically significant change in Occludin expression between the two genotypes:

Although, this does not represent a functional assay to assess epithelial barrier function, this suggests that at least at this time point, epithelial barrier integrity is not negatively affected after chronic DSS exposure by the loss of HIF1a in NKp46+ cells. This is further corroborated the assessment of mucosal bacterial invasion.

We addressed this and performed Gram staining to detect Gram positive and negative bacteria in the on colon sections from WT and HIF1a KO mice after chronic DSS exposure at endpoint (day 28). As shown below, the quantification of bacteria in the mucosa did not reveal a statistically significant difference in bacterial invasion across genotypes. Of note, we were not able to detect any bacteria in the submucosa at this given time point after chronic DSS.

In summary, this suggests that the epithelial barrier integrity is not further compromised by the loss of HIF1a in NKp46+ cells. We feel that this data could contribute to the discussion section of the manuscript but would like to leave it to the reviewers and the editor, whether these results should be included in the manuscript.

4) Please include reference to the powering of experiments and the n numbers of mice used in the Statistics section.

Our Reply:

We are not entirely sure, whether we understand the question correctly. The n number of mice for each experiment is indicated in the respective figure legend. As this is a requirement to obtain the license for the animal experiments from the authorities, the power calculation was performed with a statistician based on previous publications with murine chronic colitis models. However, we had to modify the chronic colitis model to the swiss regulations on maximum weight loss of mice during an experiment.

May 28, 2024

RE: Life Science Alliance Manuscript #LSA-2024-02593-TR

Prof. Christian Stockmann
University of Zurich
SWITZERLAND

Dear Dr. Stockmann,

Thank you for submitting your revised manuscript entitled "Expression of the transcription factor HIF-1 α in NKp46+ ILCs limits intestinal fibrosis". We would be happy to publish your paper in Life Science Alliance pending final revisions necessary to meet our formatting guidelines.

- please consider Reviewer 2's remaining comments
- please be sure that the authorship listing and order is correct and that they match in the manuscript and in our system
- please upload a clean manuscript file without highlights in the .docx file format
- the title in the manuscript and the system need to match
- If there is more than one corresponding author, please indicate so on the cover page of your manuscript. This information needs to match the information provided in the system.
- please add ORCID ID for the corresponding (and secondary corresponding) author (you should have received instructions on how to do so)
- please add the Twitter handle of your host institute/organization as well as your own or/and one of the authors in our system
- please remove label A in Supplementary Figure 3 since there are no additional figure sections
- please update the Data Availability statement with the accession information for the RNA-seq data
- under the Animal Models section, please indicate that approval was granted for the mouse work, and who granted that approval

A. FINAL FILES:

B. MANUSCRIPT ORGANIZATION AND FORMATTING:

Sincerely,

Reviewer #2 (Comments to the Authors (Required)):

We appreciate that the authors addressed all of our comments and we believe that the additional experiments significantly advanced the robustness of their scientific findings. Nevertheless, the specificity of the Cre line remains a major issue. It is important to acknowledge the potential contribution of HIF1 α deletion in NK cells to the observed phenotype, and rephrase the text of the manuscript accordingly (title and across the manuscript).

Additionally, some minor details that raise discrepancies concern Figure 2. Specifically, the representative image chosen for the LGR5 staining is not convincing, since the LGR5 signal does not correspond to DAPI+ cells. Moreover, in the quantification graph of MUC2+ cells the dots seem to represent individual measurements instead of biological replicates. If this is the case, statistical analysis is not applicable and re-analysis of the measurements is needed. Regarding the bar plot, we appreciate that the authors created this extra visualization, as we proposed. However, we have to clarify that in this graph the bars should correspond to genotypes, so that the abundances of all the different cell types identified in the single-nuclei experiment are presented in one bar per genotype. The current graph raises some concerns showing that the colon of H1 α -KO mice is enriched in immune cells compared to the WT, which is contradictory to the CD45+ cell quantification presented in Fig. 1C. Consequently, the authors should consider resolving the discrepancies mentioned above in order not to compromise the significance of their findings.

Reviewer #3 (Comments to the Authors (Required)):

The authors have addressed my concerns and comments. I am happy to recommend publication of this nice and important paper.

Referee Cross Review Comments: Reviewer two makes some additional requests to the authors some of which could have been pointed out in the initial review. However, I believe these are minor changes and should be easy to address with text changes. One issue raised by the reviewer is the exact correlation between representative images and the quantified data presented which I believe is asking a lot as one individual image which represents exactly the quantified data of large numbers of measurements can often be difficult to display.

Reviewer #2 (Comments to the Authors (Required)):

We appreciate that the authors addressed all of our comments and we believe that the additional experiments significantly advanced the robustness of their scientific findings. Nevertheless, the specificity of the Cre line remains a major issue. It is important to acknowledge the potential contribution of HIF1 α deletion in NK cells to the observed phenotype, and rephrase the text of the manuscript accordingly (title and across the manuscript).

Additionally, some minor details that raise discrepancies concern Figure 2. Specifically, the representative image chosen for the LGR5 staining is not convincing, since the LGR5 signal does not correspond to DAPI+ cells. Moreover, in the quantification graph of MUC2+ cells the dots seem to represent individual measurements instead of biological replicates. If this is the case, statistical analysis is not applicable and re-analysis of the measurements is needed. Regarding the bar plot, we appreciate that the authors created this extra visualization, as we proposed. However, we have to clarify that in this graph the bars should correspond to genotypes, so that the abundances of all the different cell types identified in the single-nuclei experiment are presented in one bar per genotype. The current graph raises some concerns showing that the colon of HIF1 α -KO mice is enriched in immune cells compared to the WT, which is contradictory to the CD45+ cell quantification presented in Fig. 1C. Consequently, the authors should consider resolving the discrepancies mentioned above in order not to compromise the significance of their findings.

Our reply:

We would like to thank the reviewer for the helpful comments, which we have addressed as follows:

We agree with the reviewer that ILC1s include NK cells and we have rephrased the text of the manuscript accordingly to acknowledge the potential contribution of HIF1 α deletion in NK cells to the observed phenotype. However, we would like to point out that our data shows that deletion of HIF1a in NKp46+ ILCs in the gut context results in a decrease in non-NK cell ILC1s but not NK cells and in general the non-NK cell ILC1s are more abundant than NK cells in our setting (Figure 1C).

We have now chosen images for the LGR5 staining that are more representative and changed the quantification graph of MUC2+ cells according to the reviewer's request.

Finally, we changed the bar plot of the the single-nuclei sequencing experiment, so that the abundances of all the different cell types identified are presented in one bar per genotype. Concerning the discrepancy that the reviewer brings up, we would like to emphasize that the FACS analysis is more accurate in quantifying immune cells than sc-RNAsequencing. Also, the FACS analysis has been performed with a higher n.

Reviewer #3 (Comments to the Authors (Required)):

The authors have addressed my concerns and comments. I am happy to recommend publication of this nice and important paper.

Referee Cross Review Comments: Reviewer two makes some additional requests to the authors some of which could have been pointed out in the initial review. However, I believe these are minor changes and should be easy to address with text changes. One issue raised by the reviewer is the exact correlation between representative images and the quantified data presented which I believe is asking a lot as one individual image which represents exactly the quantified data of large numbers of measurements can often be difficult to display.

Our reply:

We would like to thank the reviewer and appreciate the positive reception of our revised manuscript.

June 5, 2024

RE: Life Science Alliance Manuscript #LSA-2024-02593-TRR

Prof. Christian Stockmann
University of Zurich
Institute of Anatomy
Winterthurerstrasse 190
Zurich 8057
Switzerland

Dear Dr. Stockmann,

Thank you for submitting your Research Article entitled "The transcription factor HIF-1 α in NKp46+ ILCs limits chronic intestinal inflammation and fibrosis". It is a pleasure to let you know that your manuscript is now accepted for publication in Life Science Alliance. Congratulations on this interesting work.

DISTRIBUTION OF MATERIALS:

Again, congratulations on a very nice paper. I hope you found the review process to be constructive and are pleased with how the manuscript was handled editorially. We look forward to future exciting submissions from your lab.

Sincerely,
